# Dengue vaccine acceptability in Peru: A mixed-methods study in two dengue-endemic Peruvian cities

Jorge L. Cañari-Casaño[1☉], Emma B. Ortega[2☉], Alfonso S. Vizcarra[3],
Roberto Camizan-Castro[4], E. Jennifer Ríos López[3], Jhonny J. Córdova López[3],
Cristina Hidalgo[3], Luz M. Moyano[5], Amy C. Morrison[6], Valerie A. Paz-Soldan[1,2]* 

1 One Health Unit, School of Public Health and Administration, Universidad Peruana Cayetano Heredia, Lima, Peru, 2 Department of Tropical Medicine and Infectious Disease, Tulane University Celia Scott Weatherhead School of Public Health and Tropical Medicine, New Orleans, Louisiana, United States of America, 3 Behavioral Sciences Research Unit, Asociación Benéfica PRISMA, Iquitos, Peru, 4 Center for Global Health, Tumbes Facilities, Universidad Peruana Cayetano Heredia, Puerto el Cura Pizarro, Tumbes, Peru, 5 School of Medicine, Faculty of Health Sciences, Universidad Nacional de Tumbes, Tumbes, Peru, 6 Department of Pathology, Microbiology, and Immunology, School of Veterinary Medicine, University of California, Davis, California, United States of America

☉ Contributed equally to this work.
* vpazsold@tulane.edu

## Abstract

### Introduction

Dengue poses a major public health challenge in Peru, with Piura and Iquitos experiencing recurrent outbreaks and limited control options. A dengue vaccine could complement current vector control strategies and reduce transmission, yet community perceptions and barriers to uptake in pre-implementation settings remain poorly understood.

### Methodology/Principal Findings

Our mixed-methods study design was conducted in Piura and Iquitos, guided by the 5C's Framework of Vaccine Hesitancy. Sixteen focus group discussions (n = 147) explored acceptability and concerns regarding a hypothetical future dengue vaccine; a subsequent survey (n = 883) quantified vaccine acceptability. Dengue vaccine acceptability was operationalized using a theory-driven, multi-item outcome contrasting individuals willing to vaccinate ("acceptors") with those unsure about accepting vaccination ("unsure"). Multivariable logistic regression examined factors associated with vaccine uncertainty.

### Results

Using the multi-item, theory-driven classification, most (81.9%) survey participants were classified as "acceptors", 14.5% were "unsure", and 3.6% would

**Data availability statement:** Survey data analyzed in this study is available at: https://doi.org/10.5281/zenodo.19645198. Focus group transcripts were de-identified, but may still contain information that could potentially be used to identify participants; hence, numerous representative quotes for all themes are presented in SM4 and the convergence/divergence matrix is in SM5.

**Funding:** VAPS supported in part by a research grant from Investigator-Initiated Studies Program of Merck Sharp & Dohme LLC: Merck Investigator Studies Program (MISP) Clinical Research 60875. The opinions expressed in this paper are those of the authors and do not necessarily represent those of Merck Sharp & Dohme LLC. The funders had no role in study design, data collection and analysis, decision to publish, or preparation of the manuscript.

**Competing interests:** The authors have declared that no competing interests exist.

refuse all vaccines ("refusers"). Qualitative findings underscored the need for clear information on efficacy, eligibility, and side effects, provided by trusted health professionals. In multivariable analysis, compared to acceptors, the "unsures" had more negative views towards COVID-19 vaccination (OR 2.17), higher technical or higher education (OR 3.33), and reduced confidence due to the rapid COVID-19 vaccine development (OR 2.17). The "unsures" also expressed less trust in vaccine benefits (OR 0.25), less willingness to pay for a dengue vaccine (OR 0.40), and lower knowledge of dengue transmission (OR 0.45) compared to the acceptors.

## Conclusions

Although most participants expressed willingness to receive a future dengue vaccine, vaccine uncertainty was primarily shaped by factors related to confidence, convenience, and communication. Mistrust in vaccines in general or residual mistrust linked to COVID-19 vaccination experiences, and concerns about vaccine development may hinder acceptance. Tailored communication, engaging trusted local leaders, and ensuring easy access are critical for successful dengue vaccination campaigns in these endemic regions.

### Author summary

Dengue is a mosquito-borne disease that has increased substantially globally in recent years and remains a major public health problem in Peru. Cities such as Iquitos in the Amazon rainforest and Piura in the northern coast experience dengue outbreaks every year. Current dengue control efforts focus on vector control activities. New dengue vaccines offer an additional tool to reduce dengue, yet their success depends on whether communities are willing and able to receive them. In this study conducted as a baseline for future vaccine implementation, we examined community attitudes towards a future dengue vaccine in two dengue-endemic cities in Peru using focus group discussions and household surveys. Most participants expressed willingness to receive a dengue vaccine. However, uncertainty about vaccination was shaped primarily by three factors: confidence in the vaccine, convenience of access, and effective and targeted communication. Participants wanted clear and consistent information about vaccine safety, side effects, eligibility, and the number of doses required, preferably delivered by trusted health professionals. They also emphasized the importance of making vaccines easy to access through locally appropriate outreach strategies. Dengue vaccination programs should prioritize transparent communication, build public trust, and ensure convenient access for its diverse communities to maximize vaccine uptake.

## Introduction

Dengue is the most rapidly advancing vector-borne disease globally, causing more human morbidity and mortality world-wide than any other arthropod-borne virus [1–5]. In Peru during 2023 and 2024, record dengue outbreak numbers were the worst ever reported, with 256,641 reported cases in 2023 and 271,531 in 2024 [6]. The outbreak affected endemic areas, such as the states of Loreto (northern Amazon rainforest) and Piura (north coast), which have had the highest burden historically, but also areas that previously had low transmission rates, like the capital city of Lima [6].

Vector control has traditionally been the primary public health strategy for the prevention and control of dengue, focused on eliminating larval habitats through source reduction (i.e., container removal) and larviciding to reduce *Aedes aegypti* and *Ae. albopictus* vectors, as well as using emergency measures with chemical pesticides (fumigation) to kill adult mosquitoes and mitigate dengue virus (DENV) transmission [7,8]. These strategies are challenging and costly to implement effectively, as are other novel vector control measures that have been recently implemented (e.g., Wolbachia, release of sterile insect (SIT), spatial repellents, etc.) [8]. A dengue vaccine could complement these efforts by increasing immunity of human populations in at-risk areas which increase entomological thresholds (number of mosquitoes needed) to sustain *Aedes*-borne virus (ABV) transmission.

For decades, vaccine development for dengue has faced technical and safety challenges [9,10]. Currently, three vaccine candidates are either licensed or in advanced phase III clinical trials [9]. Dengvaxia (CYD-TDV, Sanofi Pasteur), partly tested in Piura, Peru, during phase II [11] has recently stopped production due to a lack of demand [12]. Recently, Qdenga (TAK003), has been approved in 40 countries and is available in 27 [13], including Peru, despite the company's withdrawal of its Biologics License Application (BLA) from the U.S. FDA [14]. The Peruvian government began roll-out of the Qdenga vaccine in four dengue endemic regions of Peru in November 2024, including the states of Loreto (where Iquitos is the capital) and Piura. A third vaccine, the Butantan DV, developed by the Brazilian Butantan Institute and U.S. National Institutes of Health (NIH) is next in line for licensure. Phase III trials for the Butantan-DV vaccine have been completed [15], and vaccination campaigns began in February 2026 in Brazil [16] following its approval. Merck, Sharp & Dohme (MSD, or "Merck" in the US) has also initiated Phase III of a single-dose V181 formulation [9].

Vaccinations are among the greatest public health advancements, significantly reducing the global burden of infectious diseases and childhood illnesses [17]. Attitudes toward vaccines, however, vary along a spectrum—from acceptance to refusal—with a gradient of hesitancy in between. Vaccine hesitancy reflects uncertainty driven by risk perception, concerns about safety, and mistrust of scientific institutions, many of which are related to access and knowledge [18]. Mis-information during the COVID-19 pandemic contributed to a decrease in vaccine uptake globally but also contributed to hesitancy and push-back towards other vaccines, particularly in the Global North [19]. Although vaccine hesitancy remains less pronounced in Peru and other dengue endemic countries compared to the United States, as dengue vaccines are rolled out [18], a clear assessment of issues associated with vaccine hesitancy and acceptability at both the community and health systems level is urgently needed.

Our study used a mixed-method approach to assess acceptability towards and barriers to uptake of a hypothetical dengue vaccine in order to propose strategies for future dengue vaccine deployment in Peru – a pre-implementation baseline that could be used to guide future rollouts. At the time our study was conducted, no dengue vaccine was commercially available in Peru, so participant experiences with COVID-19 vaccination efforts in Peru were used to provide a concrete and tangible example for understanding public perceptions and actual behaviors. Since then, the Qdenga vaccine was rolled out in 2024 in Iquitos and Piura (the two sites of this study), and a manuscript is in progress about post-implementation barriers and how these compared to pre-implementation findings. To analyze and structure our findings, we employed the '5Cs' Framework of Vaccine Hesitancy, initially proposed by the Strategic Advisory Group of Experts on Immunization (SAGE) [20], and later updated by Razai et al. [21]. This model identifies five key factors influencing vaccine hesitancy: confidence, complacency, convenience, communication, and context. Additionally, for our survey findings, we used concepts from the vaccine hesitancy continuum, focusing on three sub-groups: those who accept all vaccines ("acceptors"),

those on the continuum of vaccine hesitancy (from here on "unsure"), to the vaccine refusers ("refusers", including deniers) [20,22]. We present findings from focus group discussions and surveys conducted in two dengue-endemic Peruvian communities, exploring pre-implementation acceptability of a dengue vaccine.

## Methods

### Ethics statement

Our study protocol was approved by the Institutional Review Boards (IRBs) of PRISMA (CE0882.21) providing local Peruvian approval and the Tulane School of Public Health and Tropical Medicine (2021–1749). The University of California, Davis IRB provided an exemption. The protocol was also reviewed and approved by the Loreto Regional Health Department which oversees health research in Iquitos. Written consent was obtained from all participants prior to the initiation of focus group discussions and to surveys.

### Study design

Our mixed-methods study started with focus group discussions (FGDs) to identify barriers to community acceptability of a dengue vaccine and motivators for uptake based on the 5C's Framework of Vaccine Hesitancy [21]. Informed by the focus group discussions, we created and administered a quantitative survey to assess attitudes toward a hypothetical dengue vaccine and to identify factors associated with being "unsure" about getting vaccinated (includes individuals who might accept vaccines but have some doubts, to those who might delay their acceptance or refuse some vaccines but not all; basically, individuals who could be "moved" on the vaccine hesitancy continuum to accept a vaccine), compared with "acceptors" to vaccination.

### Setting

We worked in two regions in Peru characterized by high dengue transmission and severe outbreaks: the state of Piura in northwestern coastal Peru and Iquitos city located in the northeastern Peruvian Amazon basin. Piura reported the highest number of dengue cases in the country over the past 15 years (111,889 cases from 2007 to 2022) and ranks fifth in cumulative incidence of dengue with warning signs and severe dengue (722 cases per 100,000 inhabitants) [23]. In 2017, the El Niño phenomenon led to increased temperatures, rainfall, and flooding, triggering a major outbreak that accounted for 65% of all reported cases nationwide that year (68,290 cases) [24]. In 2023 and 2024, Piura became the epicenter of the largest dengue outbreaks in Peru, with 79,304 and 33,612 cases, respectively. In contrast, Iquitos bears the highest dengue burden in the Loreto region [6], which ranks second nationally in cumulative incidence of dengue with warning signs and severe dengue (1,518 cases per 100,000 inhabitants) from 2007 to 2022 [23]. In 2023 and 2024, dengue also spread to Lima, southern regions, and rural regions of Loreto; all historically considered low-incidence areas [6].

We started our study in Iquitos, the capital of Loreto Department, with a population of approximately 400,000 residents; surrounded on three sides by the Itaya, Nanay, and Amazon rivers, it is accessible only by boat or plane, forming a socio-epidemiological island [25]. The city is divided into four districts, each of which also have rural sections along the Iquitos-Nauta highway and adjacent rivers. Additionally, the city has been a dengue research hub since the early 1990s, contributing significantly to scientific understanding of dengue and its transmission and prevention, including the spatiotemporal dynamics, behavior, and genetic structure of the *Aedes aegypti* vector [26–34]; the community-level understanding and lived experience of dengue illness [35,36]; the evaluation of targeted and city-wide vector control strategies [37–41]; the longitudinal and interepidemic transmission patterns of dengue virus [42]; the feasibility of implementing community-based surveillance tools for febrile disease detection [43–45]; and the role of human mobility in shaping transmission risk and spatial spread [46–51]; among other relevant contributions to the field [52,53]. Based on our studies, dengue virus activity remains elevated from August to April, peaking between November and January [41,54]. From

late 2010, until late 2019, DENV-2 was the dominant serotype; thereafter, DENV-1 emerged, triggering an outbreak that was overshadowed by the COVID-19 pandemic [55]. According to the Peruvian National Institute of Statistics (INEI), 30% of the urban population in the Amazon rainforest has at least one unmet basic need, and 18.2% lives in poverty [56]. Despite these challenges, literacy rates remain relatively high, ranging from 88% to 92% [25]. Our study was carried out in the Punchana and Iquitos districts of Iquitos city, selected due to previous participation (half of the sample) in dengue research projects.

We then moved to Piura in the northern tropical coastal region of Peru, which has shown a markedly different pattern of dengue virus behavior compared to Iquitos. In the state of Piura, with an estimated population of 1,856,809 residents [25], we worked in Chulucanas and La Matanza districts (Morropón Province) and La Unión district (Piura Province). These sites were selected due to their participation (also in half of the sample) in previous research on the tetravalent Dengvaxia vaccine [11].

### Data collection instruments

To facilitate discussion in the FGDs, we used the recent and tangible experience with the COVID-19 vaccination as a starting point to examine acceptance towards a potential (but at the time, still hypothetical) dengue vaccine, as participants had all recently been engaged in this COVID-19 national vaccination program. Participants were then asked to envision a scenario where the Ministry of Health had approved a dengue vaccine and addressed three key questions: 1) What information would you need to accept vaccination? 2) How can public confidence in the vaccine be improved? 3) What is the best way to distribute a new vaccine quickly in your community? The Iquitos study team developed and piloted the FGD guides in Spanish [S1 File].

A survey was then developed using the preliminary results from the FGDs [S1 File]. The main outcome of interest was dengue vaccine acceptability, but at the time of data collection, no validated instrument was available to specifically measure dengue vaccination acceptability, particularly in pre-implementation contexts. We therefore adapted the Oxford COVID-19 Vaccine Hesitancy Scale [57], a tool designed to capture gradients of vaccine acceptance, uncertainty, and refusal prior to large-scale vaccine rollout. This framework is appropriate for dengue vaccination given the strong conceptual parallels between both scenarios, including evaluation of a novel vaccine under conditions of incomplete information, uncertainty regarding safety and effectiveness, widespread circulation of risk-related narratives, and decision-making in advance of routine implementation.

The seven Oxford hesitancy items were linguistically and contextually adapted to the dengue setting and administered to all participants [see details of process in S2 Text, S3 Text, S4 Text, S5 Text]. Following the analytical recommendations proposed by Freeman et al. [57], and to account for a minor modification introduced during pilot testing of one item, dengue vaccine acceptability was operationalized using four complementary outcome specifications: (i) a latent continuous hesitancy construct estimated using confirmatory factor analysis; (ii) an observed continuous mean hesitancy score; (iii) a three-level categorical outcome (*"acceptors" to vaccination*, "*unsure" about getting vaccinated*, *and "refusers"*); and (iv) a dichotomous outcome contrasting acceptors versus unsure about getting vaccinated (since the literature suggests focusing on moving those in the unsure category to the acceptor category [58,59]).

The survey also covered sociodemographic characteristics; experiences with COVID-19 and dengue; COVID-19 vaccination use and perceptions; vaccine attitudes (VAX scale and 5Cs Framework); participation in vector control activities; and knowledge, attitudes, and practices related to dengue prevention. Correct knowledge of dengue transmission was defined as identifying the bite of an infected mosquito as the mode of transmission. The Vaccine Attitudes Examination (VAX) scale, previously validated in Spanish, measured four constructs: "1) mistrust of vaccine benefit, 2) worries about unforeseen future effects, 3) concerns about commercial profiteering, and 4) preference for natural immunity" [60]. After piloting this instrument with 20 individuals, the response format was adapted from a 7-point to a 5-point Likert scale for comprehension, tested for internal consistency (Cronbach's $\alpha = 0.73$; standardized $\alpha = 0.72$). The survey was developed,

collected, and managed using REDCap (version 14.0.8) electronic data capture tools [61,62] and applied by our team on tablets [S1 File].

In the main manuscript, descriptive analyses are presented using the three-level categorical outcome to preserve meaningful gradients of vaccine acceptability. However, multivariable regression analyses focus on the dichotomous outcome, selected to identify determinants of vaccine uncertainty among individuals *potentially amenable* to intervention. Detailed descriptions of outcome construction and sensitivity analyses across all specifications are included [S2 Text, S3 Text, S4 Text, S5 Text].

### Recruitment and procedures

Both qualitative (FGDs) and quantitative (survey) components of our study were stratified by recruiting from intervention and control areas. In the city of Iquitos, the intervention group was defined as clusters (grouped blocks) that had previously participated in a vector control trial [38] whereas the control groups were selected from clusters where residents had no prior exposure to community-based research studies conducted by our group. Both the intervention and control groups were in Iquitos and Punchana districts of the city. We replicated this process in Piura, with the intervention group defined as districts where the Dengvaxia vaccine trial took place, located in the Chulucanas and La Matanza districts [11]. Each intervention group was matched to a control area in La Unión district, which was not involved in the vaccine trial. We hypothesized that a community's previous interaction with dengue research teams – and in Piura, a dengue vaccine trial – might influence responses, so we wanted to ensure some participants were selected from "intervened" areas.

We recruited FGD participants from a 5x5 block radius through door-to-door convenience sampling one day prior to the FGDs, limiting participation to two participants per block. Eligible participants were adults (>20 years old) who made household decisions, were mentally competent to provide consent, and met the age and gender criteria for the selected FGD. The research team returned 1–2 hours before each FGD to assist with transportation to the site. The FGD team was led by a senior qualitative researcher (VPS) and three research assistants: one co-facilitated (ASVS), one recorded participant feedback on butcher paper to facilitate discussion (EJRL in Iquitos, CC in Piura), and the third took notes (JLCC). We conducted 16 focus group discussions (FGDs) between February and March 2022 to reach saturation—eight in Iquitos, followed by eight in Piura. In both study sites, four FGDs were conducted with residents from intervention areas and four with residents from control areas [38]. We stratified all FGDs by gender (female or male) and age (20–39 or 40–60 years). All FGDs were audio-recorded.

For the quantitative phase, a base sample of ≥400 participants per study site was defined by selecting city blocks randomly, followed by systematic door to door recruitment of subjects with a maximum number of enrollees per block. Only one participant per household was permitted (see below). This target ensured stable site-specific estimates, adequate precision for a binary outcome under a conservative assumption (±5% margin of error at 95% confidence), and feasibility for door-to-door household surveys. We used random sampling to select blocks in both the intervention and control areas to target recruitment in every household within those blocks. Recruitment was door to door until target enrollment of eligible adult participants (aged 20–60) who were household decision makers was reached. ASVS oversaw training and field monitoring; JLCC managed quality control as data was entered on tablets and uploaded at least twice a day. Before starting data collection, the interviewers were trained in survey administration and research ethics. The surveys were conducted in August 2022 in Iquitos and a month later, in September, in Piura.

### Data analysis

Audio recordings of our FGDs were transcribed into Spanish using the Trint platform [63] and our research team conducted quality control (editing, verification, de-identification) of the transcripts within the Trint software [63]. We conducted thematic analysis using the 5C's – confidence, complacency, convenience, communication, and context – as deductive

Neglected Tropical
Diseases

PLOS

codes, and emerging themes were added as subcodes inductively. Data from the FGDs was managed, coded, and analyzed using Dedoose software (Version 9.0.107) [64].

For our quantitative survey, dengue vaccine hesitancy was operationalized using multiple outcome specifications, as detailed above. Descriptive analyses were conducted using the three-category outcome (vaccination "acceptors", "unsure" about getting vaccinated, and vaccination "refusers") to characterize participant attributes and response distributions across groups that reflect differing positions along the acceptance–hesitancy continuum. Categorical variables were summarized using proportions and compared using chi-square tests.

For multivariable analyses, the primary outcome was a dichotomous dengue vaccine hesitancy variable contrasting "acceptors" versus with those who were "unsure". This dichotomy was chosen to identify determinants of vaccine hesitancy among individuals who had not expressed unequivocal refusal that might provide insights about moving them from the "unsure" to the "acceptor" category. Multivariable logistic regression models were estimated to assess the association between predictors derived from the 5C's framework and the likelihood of being unsure about vaccination. Adjusted odds ratios (ORs) and 95% confidence intervals were reported.

Selection of predictors for the final multivariable model followed a structured, theory-driven approach grounded in the 5C's framework. All candidate variables were classified a priori into the five conceptual domains (Confidence, Complacency, Convenience, Communication, and Context). Within each domain, multicollinearity was assessed using variance inflation factors (VIF) and overlapping or highly correlated variables were evaluated to guide retention or exclusion. Crude and domain-specific adjusted models were examined to assess the stability of effect direction and relative magnitude within each domain. In addition to domain-specific predictors, a core set of a priori sociodemographic covariates (sex, age group, education level, and occupation) was included in the final multivariable model to control for potential confounding. These covariates were retained irrespective of statistical significance. Final inclusion of predictors in the multivariable model was guided by conceptual relevance, absence of problematic multicollinearity, consistency of effects across domain-specific analyses, and model parsimony. Variables were not selected solely based on statistical significance; conceptually important predictors were retained even when not statistically significant to ensure adequate representation of key dimensions of vaccine hesitancy. All variables evaluated within each domain, the set of predictors retained in the final model, and the results of multicollinearity diagnostics are presented [S1 Text]. Additionally, sensitivity analyses were conducted using alternative outcome specifications (three-level categorical, latent continuous, and observed continuous outcomes), applying the same predictor set [S4 Text].

Data was cleaned using Stata 15.0 (Stata Corp., College Station, Texas, United States), whereas analysis and graphics were created in R version (Version 4.4.3; R Foundation for Statistical Computing).

## Results

### Study population

A total of 147 individuals participated in the 16 FGDs: 72 from Iquitos and 75 from Piura. Over half of participants were female (53%), and 52% were aged 20–39. In 15 of the 16 FGDs, at least one participant had experienced dengue. Most participants had received at least one dose of a COVID-19 vaccine [Table 1]. The full table of quotes in English and the original Spanish can be found in the supplemental files [S1 Table].

A total of 883 of 1,240 invited participants (71%) were enrolled in the survey (see Fig 1). Of the survey respondents, 122 (13.8%) were excluded from the final multivariable analysis – 42 from Piura (9.3%) and 80 from Iquitos (18.5%) – due to classification as "refusers" of our outcome variable of dengue vaccine acceptability or missing covariate data under a complete-case approach. The final regression models were estimated using the 761 participants. Nearly half of participants were aged 40–60 (48.7%), the majority were women (71.1%), and around 83.5% had completed secondary education or higher [Table 2].

**Table 1. Focus Group Discussion Sample Characteristics.**

| Site | Gender | Age Group | Number of Participants | Participants who have had dengue | Participants with at least ≥1 COVID-19 Vaccine Dose |
|---|---|---|---|---|---|
| Iquitos (n=72) (8 FGDs) | Male | 20–39 | 16 | 6 | Not recorded* |
| | | 40–60 | 17 | 5 | 17 |
| | Female | 20–39 | 19 | 1 | 18 |
| | | 40–60 | 20 | 11 | 19 |
| Piura (n=75) (8 FGDs) | Male | 20–39 | 23 | 7 | 23 |
| | | 40–60 | 14 | 2 | 14 |
| | Female | 20–39 | 19 | 7 | 19 |
| | | 40–60 | 19 | 2 | 19 |
| TOTAL | | | 147 | 44 | 138 |

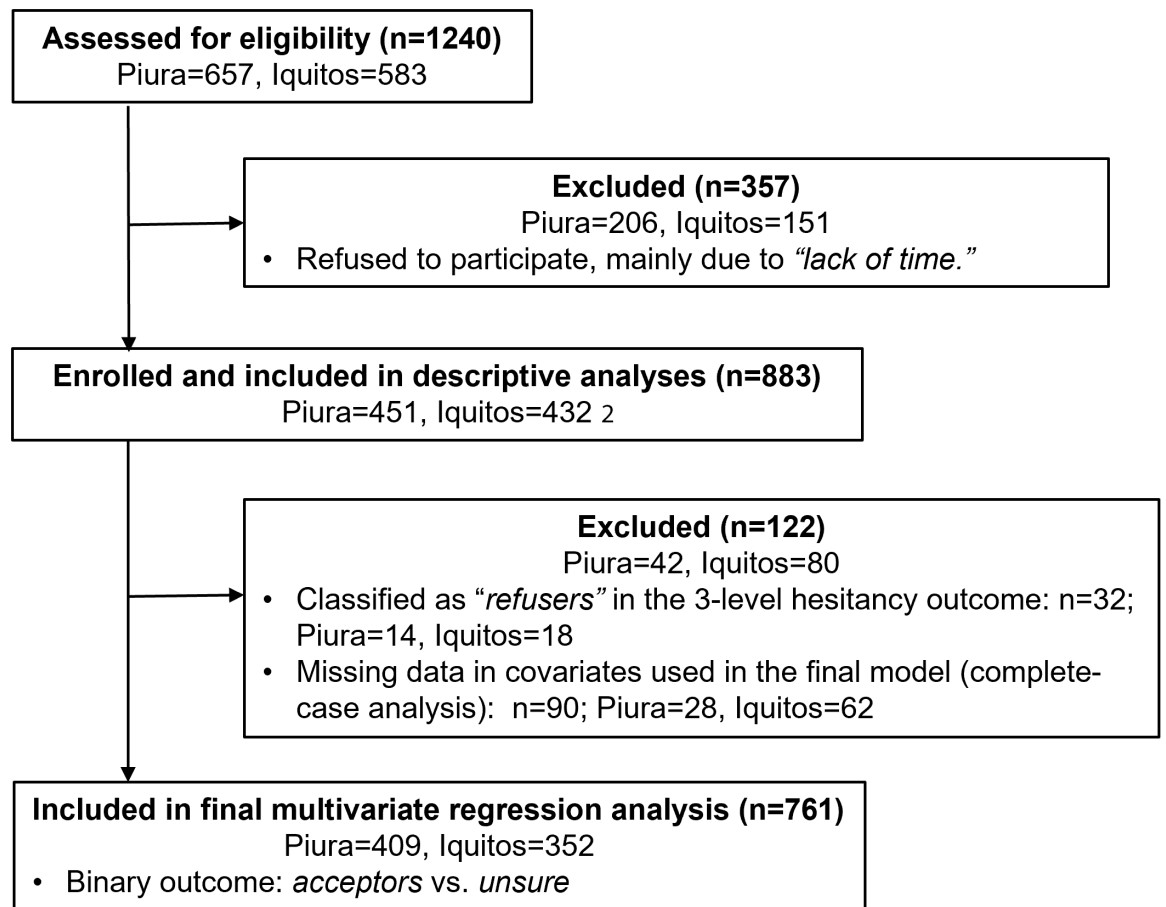

**Fig 1. Flow diagram of survey participant enrollment and inclusion in data analysis.** A characterization of the 90 participants excluded from the multivariable model due to missing covariate data are available in S6 Text.

**Table 2. Sociodemographic Characteristics of Survey Participants.**

| Characteristic | Overall N = 883[1] | Iquitos N = 432[1] | Piura N = 451[1] | p-value[2] |
|---|---|---|---|---|
| **Residence area** | | | | <0.001 |
| Urban | 81.7% (721/883) | 100.0% (432/432) | 64.1% (289/451) | |
| Peri-urban | 18.3% (162/883) | 0.0% (0/432) | 35.9% (162/451) | |
| **Gender** | | | | <0.001 |
| Female | 71.1% (628/883) | 63.2% (273/432) | 78.7% (355/451) | |
| Male | 28.9% (255/883) | 36.8% (159/432) | 21.3% (96/451) | |
| **Age** | | | | 0.13 |
| 20 to 39 years old | 51.3% (453/883) | 53.9% (233/432) | 48.8% (220/451) | |
| 40 to 60 years old | 48.7% (430/883) | 46.1% (199/432) | 51.2% (231/451) | |
| **Education Level** | | | | <0.001 |
| Elementary school | 16.4% (145/883) | 3.5% (15/432) | 28.8% (130/451) | |
| High school | 45.6% (403/883) | 46.5% (201/432) | 44.8% (202/451) | |
| Technical and/or higher | 37.9% (335/883) | 50.0% (216/432) | 26.4% (119/451) | |
| **Occupation** | | | | <0.001 |
| Homemaker | 44.7% (392/877) | 32.6% (139/427) | 56.2% (253/450) | |
| Skilled/professional | 32.8% (288/877) | 45.0% (192/427) | 21.3% (96/450) | |
| Manual Laborer | 22.5% (197/877) | 22.5% (96/427) | 22.4% (101/450) | |
| **Household size (total number of residents)** | | | | <0.001 |
| 1 to 3 people | 22.3% (197/883) | 17.1% (74/432) | 27.3% (123/451) | |
| 4 to 6 people | 53.8% (475/883) | 48.6% (210/432) | 58.8% (265/451) | |
| 7 to more people | 23.9% (211/883) | 34.3% (148/432) | 14.0% (63/451) | |
| **Close experience with severe dengue** | 15.2% (134/883) | 19.9% (86/432) | 10.6% (48/451) | <0.001 |
| **Dengue vaccine hesitancy** | | | | 0.5 |
| Vaccination "acceptors" | 81.9% (723/883) | 80.3% (347/432) | 83.4% (376/451) | |
| Unsure to get vaccinated | 14.5% (128/883) | 15.5% (67/432) | 13.5% (61/451) | |
| Vaccination refusers | 3.6% (32/883) | 4.2% (18/432) | 3.1% (14/451) | |

[1]% (n/N)

[2]Pearson's Chi-squared test

### 5C's framework

Below we present a summary of both qualitative (relevant quotes) and survey findings, organized under the 5C's framework.

### Confidence

Both qualitative and quantitative data indicated generally positive attitudes towards a potential dengue vaccine, and the scales showed relatively low vaccine hesitancy [Tables 2 and 3]. The following statements from FGD reflected this sentiment: *"We assume that a vaccine is good for your health. Who wouldn't want to get it? But as long as we know what symptoms [side effects] we will get. Right?"* (FG1). Survey results were consistent with FGDs findings, with 81.9% (723) of respondents indicating they were willing to receive a new dengue vaccine when available [Table 2]. Two main themes emerged in FGDs regarding confidence about a new vaccine: 1) the need for clear information about the vaccine to increase people's confidence in it, including information on its efficacy, number of doses required for protection, and possible side effects, and 2) vaccine information should come from trusted institutions, naming the Ministry of Health and health professionals as such [Table 3].

Most survey participants highly trusted the benefits of vaccines (69.6%) [Table 4]. There was mild preoccupation about potential future vaccine effects both among FGD and survey participants. FGD participants said they wanted

**Table 3. Themes Identified in the FGDs and Representative Quotes.**

| Themes | Illustrative Quotes |
|---|---|
| **Confidence** | |
| Vaccine side effects | *We imagine that it's a vaccine which is good for your health. Who wouldn't want to get it? But as long as they end up knowing what symptoms we're going to have. Right? (FG1)*<br>*And the fear of a reaction to the vaccine. (FG1)* |
| Vaccine eligibility | *Have they tested it on a person who's had dengue, including a severe case? (FG9)*<br>*What problems would it cause in a pregnant woman, or a diabetic person, or for any other disease a person may have? (FG10)* |
| Evidence from others | *First, like I said, wait for others to get vaccinated, and see what reactions they have. (FG5)* |
| Number of doses | *Yes, because he already has three vaccines, three doses, a vaccine is never three doses, it's only two. (FG1)* |
| Science | *Because I always say that vaccines protect against the bad that's happening in these times. (FG1)*<br>*To protect ourselves from the disease and logically we have to get vaccinated, because if we are not vaccinated we are at risk for the disease (FG2)* |
| Immunity | *Depending on the guarantees they would give me- if it's immunization for life, then I would get it. (FG2)* |
| Experience with Dengue | *Well, as someone from Loreto, I've had dengue at some point. (FG2)*<br>*I think the dengue vaccine is something that is highly anticipated and I think this event would be a boom. (FG2)*<br>*Because now it's different, because the diseases are evolving and they are no longer the same, I think. Now the disease is stronger, well that's what I think. (FG13)* |
| Local trusted sources | *Well, here in Unión, we honestly don't trust the authorities. What they promise are lies. Maybe they were telling me something that isn't true. (FG11)* |
| National sources | *The first thing I would do is find out if the Ministry of Health has approved it. (FG2)* |
| **Complacency** | |
| Self-perceived risk from past experience | *But when you get dengue hemorrhagic fever, you start to worry. That's when you start to realize it's not a joke, with dengue. (FG13)*<br>*Dengue has been around for many years, but what did we do? What did we use to cure dengue? Pure bitters, and we still use them. (FG13)* |
| Protecting the health of children | *It's a responsibility, as a parent, to get them vaccinated, so that they grow up healthy and strong with their childhood vaccines. (FG11)*<br>*Well, to me, they should vaccinate the little ones first, to avoid that dengue disease. (FG14)* |
| Work/ Travel and Policies | *Because to enroll in school they will force you to get the children vaccinated. (FG4)*<br>*To play football they are asking for the vaccine, if you don't have it, you can't play. (FG5)* |
| Low self-perceived risk | *It's never happened to me. I mean, I've never gotten sick enough to call a doctor. (FG6)*<br>*I'm not afraid because you just get a mild dengue infection, right? (FG13)* |
| Religious views | *For the simple reason that it says it is the sign of the beast. (FG1)* |
| Vaccine effect on vector control | *As for me, what I want to know is if I get vaccinated, then I no longer need to kill my mosquitoes? (FG2)*<br>*For dengue, we should put on repellent or put [a repellent device on wall] or use a mosquito coil, right? To protect ourselves, we should continue doing this, even if we've been vaccinated? (FG10)* |
| **Convenience** | |
| Long lines | *People got up early, brought their chairs, put them in front of the line, and they sold their spot for 10 soles or 20 soles. (FG1)* |
| Costs | *Generally, vaccines are provided by the government. I don't know of a vaccine that we can buy. (FG2)*<br>*I prefer to save my family, I can pay whatever they tell me to. 50, 20 soles, I'll pay it. It's to protect us, to save our lives. (FG11)* |
| Ideas for rolling out the dengue vaccine | *And sustained awareness is very important, not just going once and then not returning. To change the mental structure of the people in the community, it [awareness building] has to be constant, you have to be with them, try different [messages]. (FG3)* |
| **Communication** | |
| Misinformation | *If you're vaccinated you would become sterile or something like that, so she hasn't had a single vaccine (FG10)* |
| Sources of information | *I don't trust the vaccine for reasons I saw on the news. I let myself be influenced by the news (FG7)*<br>*The radio station, well, the station as well. (FG10)* |

**Table 4. 5C's by dengue vaccine acceptability category: vaccination acceptors, those who are unsure, and vaccination refusers.**

| 5C's Domain | Overall N=883[1] | "Acceptors" N=723[1] | Unsure about getting vaccinated N=128[1] | "Refusers" N=32[1] | p-value[2] |
|---|---|---|---|---|---|
| **CONFIDENCE** | | | | | |
| Opinion of the COVID-19 vaccine | | | | | <0.001 |
| *In favor* | 80.6% (712/883) | 85.9% (621/723) | 63.3% (81/128) | 31.3% (10/32) | |
| *Against or in doubt* | 19.4% (171/883) | 14.1% (102/723) | 36.7% (47/128) | 68.8% (22/32) | |
| Trust in vaccine benefits (Vaccination Attitudes Examination [VAX] scale construct) | | | | | <0.001 |
| *Low* | 6.1% (50/819) | 3.3% (22/665) | 15.4% (19/123) | 29.0% (9/31) | |
| *Intermediate* | 24.3% (199/819) | 21.2% (141/665) | 39.0% (48/123) | 32.3% (10/31) | |
| *High* | 69.6% (570/819) | 75.5% (502/665) | 45.5% (56/123) | 38.7% (12/31) | |
| Knowing the COVID-19 vaccine was developed very quickly reduced my confidence in getting vaccinated | 12.0% (106/883) | 9.8% (71/723) | 20.3% (26/128) | 28.1% (9/32) | <0.001 |
| Concerns about unforeseen future effects of vaccines (VAX scale construct) | | | | | 0.034 |
| *Low* | 11.0% (96/876) | 12.1% (87/719) | 3.9% (5/127) | 13.3% (4/30) | |
| *Intermediate* | 52.1% (456/876) | 51.3% (369/719) | 54.3% (69/127) | 60.0% (18/30) | |
| *High* | 37.0% (324/876) | 36.6% (263/719) | 41.7% (53/127) | 26.7% (8/30) | |
| **COMPLACENCY** | | | | | |
| Participant believes they could get dengue again if previously infected | 59.3% (523/882) | 62.2% (449/722) | 43.0% (55/128) | 59.4% (19/32) | <0.001 |
| Participant believes people can die from dengue infection | 94.4% (833/882) | 96.1% (694/722) | 88.3% (113/128) | 81.3% (26/32) | <0.001 |
| Participant believes dengue infection could lead to a loss of income | 86.3% (760/881) | 88.1% (635/721) | 78.9% (101/128) | 75.0% (24/32) | 0.003 |
| Preference for natural immunity | | | | | 0.003 |
| *Low* | 39.1% (340/869) | 40.8% (291/713) | 32.5% (41/126) | 26.7% (8/30) | |
| *Intermediate* | 33.7% (293/869) | 30.7% (219/713) | 46.8% (59/126) | 50.0% (15/30) | |
| *High* | 27.2% (236/869) | 28.5% (203/713) | 20.6% (26/126) | 23.3% (7/30) | |
| **CONVENIENCE** | | | | | |
| Willing to pay for a dengue vaccine | 40.2% (355/883) | 45.2% (327/723) | 21.1% (27/128) | 3.1% (1/32) | <0.001 |
| Willing to wait in line at dengue vaccination site | | | | | 0.018 |
| *Less than one hour* | 38.9% (343/881) | 36.8% (266/722) | 50.0% (64/128) | 41.9% (13/31) | |
| *One hour or more* | 61.1% (538/881) | 63.2% (456/722) | 50.0% (64/128) | 58.1% (18/31) | |
| | | | | | |
| **CONTEXT** | | | | | |
| Previous participation in dengue research | 47.9% (423/883) | 49.0% (354/723) | 42.2% (54/128) | 46.9% (15/32) | 0.4 |
| Close experience with severe COVID-19 | 36.0% (318/883) | 37.1% (268/723) | 28.1% (36/128) | 43.8% (14/32) | 0.10 |
| Close experience with severe dengue | 15.2% (134/883) | 15.4% (111/723) | 14.1% (18/128) | 15.6% (5/32) | >0.9 |
| COVID-19 vaccine doses | | | | | <0.001 |
| *Third or fourth doses received* | 83.5% (737/883) | 86.4% (625/723) | 71.9% (92/128) | 62.5% (20/32) | |
| *None or incomplete series* | 16.5% (146/883) | 13.6% (98/723) | 28.1% (36/128) | 37.5% (12/32) | |
| Correct knowledge of how dengue is transmitted | 88.1% (775/880) | 89.5% (645/721) | 82.8% (106/128) | 77.4% (24/31) | 0.016 |
| **COMMUNICATION** | | | | | |
| Before getting vaccinated against dengue, I would like to know about the vaccine's adverse effects | 83.1% (734/883) | 85.3% (617/723) | 78.1% (100/128) | 53.1% (17/32) | <0.001 |

*(Continued)*

**Table 4.** (Continued)

| 5C's Domain | Overall N = 883[1] | "Acceptors" N = 723[1] | Unsure about getting vaccinated N = 128[1] | "Refusers" N = 32[1] | p-value[2] |
|---|---|---|---|---|---|
| The Ministry of Health is a source for dengue vaccine information | 62.3% (550/883) | 63.9% (462/723) | 55.5% (71/128) | 53.1% (17/32) | 0.11 |
| Negative information or misinformation encountered through news media or social networks was a major factor contributing to my distrust of the COVID-19 vaccination | 42.5% (375/883) | 41.1% (297/723) | 50.0% (64/128) | 43.8% (14/32) | 0.2 |
| Health centers or local primary care facilities are a source for dengue vaccine information | 40.8% (360/883) | 44.1% (319/723) | 28.1% (36/128) | 15.6% (5/32) | <0.001 |
| Desire to know the number of doses before dengue vaccination | 33.4% (295/883) | 34.9% (252/723) | 28.9% (37/128) | 18.8% (6/32) | 0.085 |
| Before deciding whether to receive a dengue vaccine, I would like to know whether it has been officially approved by Ministry of Health or WHO | 7.4% (65/883) | 6.1% (44/723) | 13.3% (17/128) | 12.5% (4/32) | 0.008 |

[1]% (n/N)

[2]Pearson's Chi-squared test; Fisher's exact test; NA

information on the vaccine's components, duration of protection, potential side effects, and whether people in other areas or countries had already been vaccinated – or if they would be first: *"I will wait for others to get it, but I am not going to risk getting it first"* (FG11). There were also questions regarding vaccine eligibility, ranging from whether it was available to all or if there were conditions such as having had dengue before, not being pregnant or not having co-morbidities: *"What problems would it cause in a pregnant woman or a diabetic [person] or any other disease a person may have?"* (FG10).

In the vaccination attitudes examination scale (VAX), the top three statements that participants agreed with focused on trusting vaccines against severe infectious diseases (83%), feeling protected after being vaccinated (81%), and feeling safe after vaccinations (78%) [Fig 2]. In our adjusted multivariate model showed that participants who expressed doubts about the COVID-19 vaccine also expressed doubts about the dengue vaccine (OR 2.17, 95% CI 1.23-3.79) [Table 5]. Likewise, those who had reduced confidence in the COVID-19 vaccine due to its rapid development were twice as likely to express uncertainty about getting the dengue vaccine than acceptors (OR 2.17, 95% CI 1.17, 3.98) [Table 5].

### Complacency

Complacency was assessed based on a person's perceived risk of getting infected and, if infected, their perception of disease severity. Most survey participants believed dengue can lead to death (94.4%) yet there were differences between the groups, as only 81.3% of refusers and 88.3% of "unsures" believed this compared to the 96.1% of acceptors [Table 4]. The perceived severity of dengue seemed to influence FGD participants' attitudes toward vaccination, with some expressing a strong desire for protection and high interest in a future dengue vaccine, particularly to prevent severe disease: *"When you get severe dengue, then you start to worry, then you start to become more conscious that dengue is no joke!"* (FG 13). There was also recognition that one could get dengue multiple times in these discussions, worthy of prevention: *"I would get the vaccine, because… there are four types of dengue… I would fear hemorrhagic dengue the most. Besides that, dengue comes every year or when the river grows in our region. I would get it [vaccine]"* (FG3). Only 59.3% of survey participants believed you can get dengue more than once [Table 4], and in the crude multivariate model, unsure

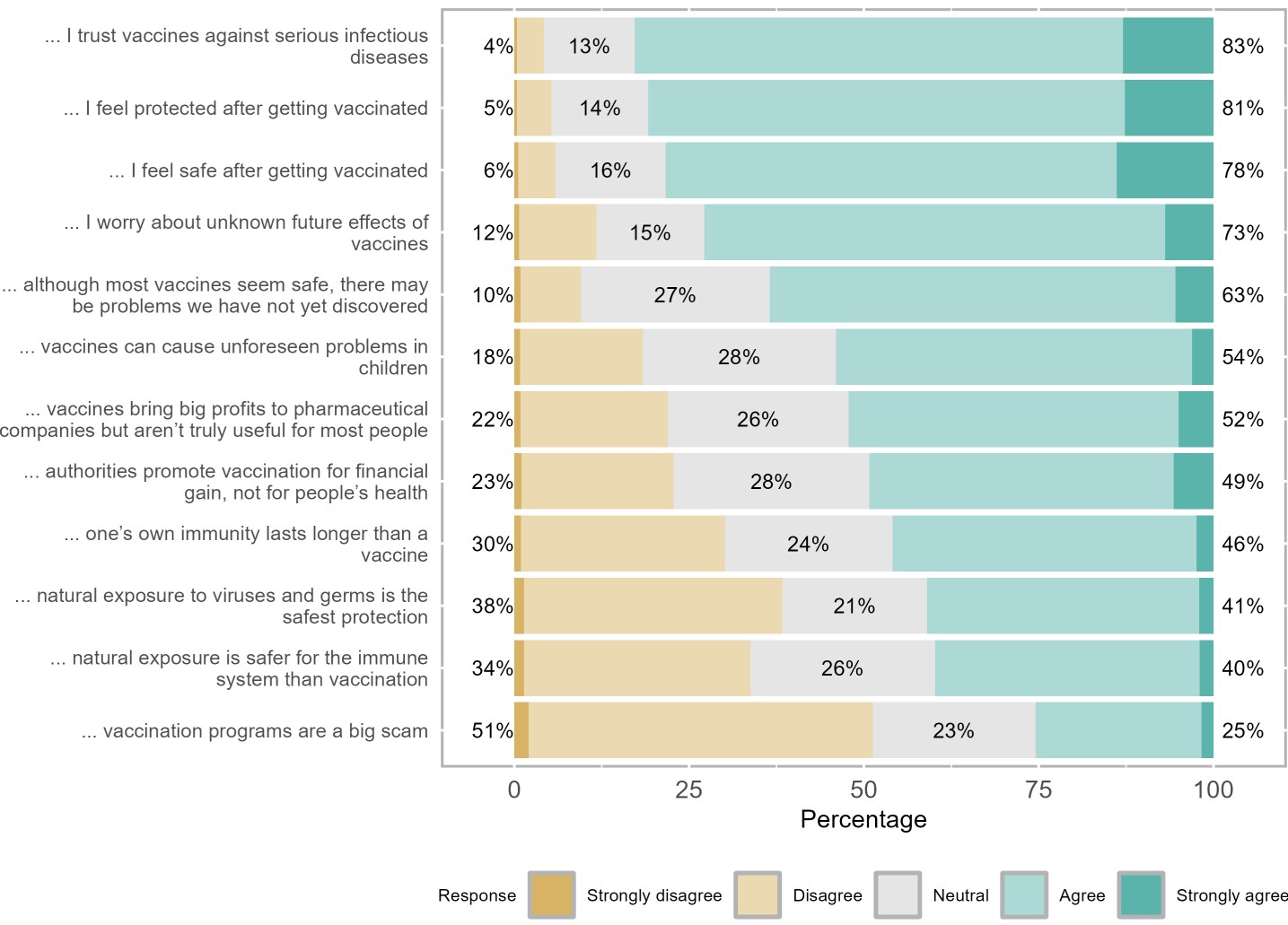

**Fig 2. Participant responses to Vaccination Attitudes Examination (VAX) Scale.** Note: Percentage on the left represents the sum of "Strongly disagree" and "Disagree" responses, whereas the percentage on the right represents the sum of "Agree" and "Strongly agree" responses".

participants were less likely than acceptors to believe they could get dengue again if previously infected, but this was not significant in the adjusted model (OR 0.71, 95% CI 0.43-1.16) [Table 5]. Mothers in the FGDs showed interest particularly in vaccinating their children, with one suggesting that vaccine eligibility should start with children. Even parents hesitant to vaccinate themselves acknowledged the importance of protecting their children as part of their parental duty. Sex was found to be a significant predictor for being unsure about getting vaccinated in the crude model but not significant in the adjusted model: men were more likely to be unsure about getting vaccinated than women (OR 1.63, 95% CI 0.91, 2.93) [Table 5].

In the FGDs, people asked whether getting vaccinated would mean they would no longer need to participate in vector control activities: *"As for me, what I want to know is if I get vaccinated, then I no longer need to kill my mosquitoes?"* (FG2).

**Table 5. Unadjusted and adjusted multivariate analysis of factors associated with being unsure about dengue vaccination, compared to acceptors (reference).**

| Characteristic | N | Unsure about vaccination (Ref: acceptors) | | | | | |
|---|---|---|---|---|---|---|---|
| | | Unadjusted model | | | Adjusted multivariable model (n = 761) | | |
| | | OR | 95% CI | p-value | OR | 95% CI | p-value |
| **CONFIDENCE** | | | | | | | |
| Opinion of the COVID-19 vaccine | 851 | | | <0.001 | | | **0.007** |
| *In favor* | | — | — | | — | — | |
| *Against or in doubt* | | 3.53 | 2.32, 5.34 | | **2.17** | **1.23, 3.79** | |
| Trust in vaccine benefits (Vaccination Attitudes Examination [VAX] scale construct) | 788 | | | <0.001 | | | **0.001** |
| *Low* | | — | — | | — | — | |
| *Intermediate* | | 0.39 | 0.20, 0.79 | | **0.52** | **0.21, 1.27** | |
| *High* | | 0.13 | 0.07, 0.25 | | **0.25** | **0.10, 0.61** | |
| Knowing the COVID-19 vaccine was developed very quickly reduced my confidence in getting vaccinated | 851 | 2.34 | 1.41, 3.80 | 0.001 | **2.17** | **1.17, 3.98** | **0.015** |
| **COMPLACENCY** | | | | | | | |
| Participant believes they could get dengue again if previously infected | 850 | 0.46 | 0.31, 0.67 | <0.001 | 0.71 | 0.43, 1.16 | 0.2 |
| **CONVENIENCE** | | | | | | | |
| Willing to pay for a dengue vaccine | 851 | 0.32 | 0.20, 0.50 | <0.001 | **0.40** | **0.23, 0.67** | **<0.001** |
| **CONTEXT** | | | | | | | |
| Study Site | 851 | | | 0.4 | | | 0.4 |
| *Iquitos* | | — | — | | — | — | |
| *Piura* | | 0.84 | 0.58, 1.22 | | 1.34 | 0.68, 2.67 | |
| Previous experience in dengue research | 851 | 0.76 | 0.52, 1.11 | 0.2 | 0.82 | 0.50, 1.35 | 0.4 |
| COVID-19 vaccine doses | 851 | | | <0.001 | | | 0.087 |
| *Third or fourth doses* | | — | — | | — | — | |
| *None or incomplete series* | | 2.50 | 1.59, 3.85 | | 1.66 | 0.93, 2.93 | |
| Correct knowledge of how dengue is transmitted | 849 | 0.57 | 0.34, 0.97 | 0.039 | **0.45** | **0.23, 0.93** | **0.030** |
| **COMMUNICATION** | | | | | | | |
| Health centers or local primary care facilities are a source for dengue vaccine information | 851 | 0.50 | 0.32, 0.74 | <0.001 | 0.55 | 0.29, 1.02 | 0.057 |
| **SOCIODEMOGRAPHIC** | | | | | | | |
| Gender | 851 | | | <0.001 | | | 0.10 |
| *Female* | | — | — | | — | — | |
| *Male* | | 2.11 | 1.43, 3.11 | | 1.63 | 0.91, 2.93 | |
| Level education | 851 | | | 0.019 | | | **0.031** |
| *Elementary school* | | — | — | | — | — | |
| *High school* | | 1.47 | 0.81, 2.84 | | **1.81** | **0.81, 4.32** | |
| *Technical and/or higher* | | 2.19 | 1.21, 4.20 | | **3.33** | **1.31, 8.97** | |

Abbreviations: CI = Confidence Interval, OR = Odds Ratio. Adjusted model complete-case N = 761.

The multivariable model was further adjusted for a set of conceptually relevant variables drawn from the 5C's framework that did not reach statistical significance and are therefore not displayed in the table but can be found in S4 Text. These included concerns about unforeseen vaccine effects (confidence); beliefs related to dengue severity, potential income loss, and preference for natural immunity (complacency); willingness to wait in line for vaccination (convenience); exposure to negative vaccine information, need for institutional endorsement, preferred communication channels, and reliance on the Ministry of Health as an information source (communication); prior close experience with severe COVID-19 or dengue (context); as well as additional sociodemographic characteristics, including age and occupation. Corresponding estimates for these variables are provided in the S4 Text.

## Convenience

Participants in FGDs reported long waiting lines as a barrier to accessing the COVID-19 vaccine; some specified queuing for hours in the morning, which was particularly challenging for individuals with work or caregiving responsibilities. Some admitted to a "known practice" of hiring others to hold their place in line, though this option was not affordable for everyone: *"Yes, some people got up very early, took their chairs and sat at the front of the line, and then would sell their spot for 10-20 soles (US$3-6)"* (FG1). However, none stated long waiting lines as a reason not to get the vaccine.

Most survey respondents believed the dengue vaccine should be free, as with the COVID-19 vaccine, with less than half (40.2%) of those surveyed reporting willingness to pay for it [Table 4]. Those unsure of getting vaccinated reported less willingness to pay for the vaccine than the acceptors (OR 0.40, 95% CI 0.23, 0.67) [Table 5]. In the FGDs, it was pointed out that the vaccine should be covered by fees paid in taxes or part of government healthcare coverage. These participants also suggested prices ranging from ~$1.40-$13.60 USD, if necessary. Some argued that the vaccine cost was justified compared to the expenses of treating severe or hemorrhagic dengue. One individual noted that even the cost of over-the-counter drugs for mild dengue could exceed the vaccine cost: *"because one might not only be in a hospital… Sometimes you will need to buy something [medication] for the headaches. For prevention, I might as well get a vaccine that costs 30 soles [~US$ 8] and then I won't have other expenses"* (FG1). Participants in FGDs also highlighted that large families would struggle to afford multiple doses and proposed sliding scale costs or discounts, such as ~$8.15 USD per person or ~$13.60 USD for two people.

## Context

Among participants who reported receiving a third or fourth dose of the COVID-19 vaccine (n = 737), 86.4% were classified as acceptors for a vaccine against dengue, compared with 71.9% among those who were unsure and 62.5% among refusers [Table 4]. Reasons given for the COVID-19 vaccine in the FGDs included having high-risk household members, community pressure to do so, and feeling sufficiently informed to decide. Some FGD participants cited obligation due to Peruvian laws requiring proof of vaccination for travel, banking, grocery shopping, or work as their main reason for getting vaccinated – with some saying it was the nudge they needed to get it.

In our study sites, all FGD and survey participants had either had dengue or knew someone who had it. Additionally, 15.2% of survey participants reported close experience either through personally having or knowing someone with severe dengue [Table 4]. Study site and previous participation in dengue research did not have an impact on survey participants' vaccine acceptability [Tables 4 and 5]. There was also a higher level of dengue vaccine hesitancy among survey participants with a technical or higher education level, a wealthier socioeconomic status, and those living in an urban area. Specifically, participants with a technical or higher education level were more likely to express being unsure about getting a dengue vaccine compared to those with a primary level education (OR 3.33, 95% CI 1.31, 8.97) [Table 5]. No differences in hesitancy were found were found between age groups, occupation of the participant, and close experience with dengue/COVID-19. Individuals who were unsure about a vaccine were also less likely than acceptors to know how dengue is transmitted (via the bite of an infected mosquito) (OR 0.45, 95% CI 0.23, 0.93) [Table 5].

Drawing on their experience with COVID-19 vaccination, FGD participants recommended establishing vaccination posts in central, frequented areas, such as city centers. Despite these suggestions, FGD and survey respondents also identified challenges in reaching certain populations, including individuals with specific religious affiliations (e.g., evangelical), indigenous communities; and, only mentioned in the survey, people living in peri-urban areas and of lower socioeconomic status [Table 6].

## Communication

At both sites, when asked about most trusted sources of communication about a new dengue vaccine, FGD participants expressed trust in their physicians, healthcare friends, and health professionals or community health workers from local facilities to endorse a vaccine. They also stated they would be more likely to get vaccinated for dengue if the Ministry of Health approved

**Table 6. Targeted Strategies for Vaccine Outreach.**

| Characteristic | Overall N = 883[1] | Acceptors N = 723[1] | Unsure to get vaccinated N = 128[1] | Refusers N = 32[1] | p-value[2] |
|---|---|---|---|---|---|
| **It would be difficult to vaccinate:** | | | | | |
| People who live outside the city | 45.3% (400/883) | 46.6% (337/723) | 43.0% (55/128) | 25.0% (8/32) | 0.045 |
| Religious or evangelical people | 38.3% (338/883) | 39.3% (284/723) | 35.9% (46/128) | 25.0% (8/32) | 0.2 |
| Indigenous people | 26.6% (235/883) | 27.2% (197/723) | 22.7% (29/128) | 28.1% (9/32) | 0.5 |
| People with low economic status | 9.2% (81/883) | 8.9% (64/723) | 9.4% (12/128) | 15.6% (5/32) | 0.4 |
| | | | | | |
| **Media to publicize the vaccine** | | | | | 0.3 |
| Radio | 17.6% (153/869) | 17.5% (125/713) | 15.2% (19/125) | 29.0% (9/31) | |
| TV | 13.8% (120/869) | 14.6% (104/713) | 11.2% (14/125) | 6.5% (2/31) | |
| Social media | 26.7% (232/869) | 26.5% (189/713) | 29.6% (37/125) | 19.4% (6/31) | |
| Loudspeaker broadcasting | 31.6% (275/869) | 31.3% (223/713) | 35.2% (44/125) | 25.8% (8/31) | |
| House-to-house notification | 10.2% (89/869) | 10.1% (72/713) | 8.8% (11/125) | 19.4% (6/31) | |

[1] % (n/N)

[2] Fisher's exact test

it. In Iquitos, one name came up in every FGD: most valued the opinion of a local physician and parish priest who had been influential and demonstrated commitment to the community at the height of the COVID-19 pandemic. In contrast, Piura participants expressed distrust in their local leaders and felt abandoned by the local government, and yet, despite these negative perceptions about health facilities in both sites, FGD participants expressed wanting approval from these same organizations and health professionals before receiving a future dengue vaccine. Participants who were unsure about getting a dengue vaccine were less likely to identify health centers or local primary care facilities as a source of dengue vaccine information than acceptors (OR 0.55, 95% CI 0.29, 1.02) [Table 5]. Non-health care sources were also mentioned: FGD participants described receiving and believing information about COVID-19 vaccines from non-healthcare sources such as radio, television (local and national programs), news channels, internet sources (including social media), religious organizations, and family and friends.

To promote a dengue vaccine, FGD and survey respondents recommended used the radio (including regional programs such as *La Voz de la Selva, Radio Loreto, Radio Saby Star*, and national programs, such as *Radio Exitosa*), TV programs, social media like Facebook, and using a truck with loudspeakers to drive around neighborhoods to share information (more commonly suggested in Piura where this method is used routinely in some communities) [Table 6]. They also suggested press releases. For healthcare engagement, participants suggested holding open forum townhalls where health professionals could educate and answer questions.

Survey data showed that 50% of those unsure and 43.8% of refusers were influenced by their experience with negative public information about the COVID-19 vaccination [Table 4]. Participants in FGDs noted that clear messaging is crucial, describing how changing COVID-19 vaccine recommendations had eroded their trust. Importantly, in most FGD, participants mentioned that health promotion messages must be tailored since sub-populations may face different barriers, specifying certain religious groups that had refused COVID-19 vaccination and indigenous populations that may need different outreach strategies due to linguistic, cultural, and geographic barriers.

## Comparison of qualitative and quantitative findings

Qualitative and quantitative findings were largely convergent [S2 Table]. Both data sources demonstrated high acceptability of a potential dengue vaccine, with confidence—shaped by trust, transparency, and prior vaccination

experiences—emerging as a central determinant. Strong convergence was also observed in the domain of communication: participants emphasized the importance of disseminating information through multiple media channels (e.g., radio, loudspeakers, social media) and from trusted sources. FGD participants also emphasized high levels of trust in health professionals, although this did not quite translate into a significant finding in the quantitative analysis. Misinformation and residual mistrust related to the COVID-19 vaccination experience emerged in both datasets, underscoring the need for clear and consistent messaging. FGDs, however, provided more detailed insight into informational needs, including safety, side effects, eligibility criteria, number of doses, and prior vaccine use. In the domain of convenience, logistical barriers such as waiting times were not described as deterrents in either FGDs or surveys, yet cost emerged as a salient factor; willingness to pay was lower among participants classified as unsure, and FGD participants generally expected vaccination to be free. For complacency, qualitative findings emphasized lived experience with dengue as a motivator, whereas survey data highlighted perceived risk and knowledge of dengue transmission as factors associated with acceptance. Finally, contextual findings were complementary: FGDs identified Indigenous and certain religious communities as potentially harder to reach, while survey analyses identified sociodemographic correlates of vaccine uncertainty.

## Discussion

As dengue vaccines are introduced in endemic settings, assessing factors that may shape uptake prior to large-scale dissemination is key for effective program design. In this mixed-methods study in two dengue endemic regions of Peru, overall acceptability of a hypothetical dengue vaccine was high with more than four-fifths of survey respondents expressing willingness to be vaccinated, consistent with findings from a meta-analysis of 19 studies in Latin America and Asia [65,66]. Although 3.6% of survey respondents indicated they would refuse vaccination under all circumstances, we focus on individuals who expressed uncertainty about getting vaccinated representing a small but programmatically important group of individuals whose concerns centered on confidence, convenience, and communication. Our findings suggest that future dengue vaccination efforts in Peru and similar settings should prioritize strategies aimed at reducing vaccine hesitancy by generating demand through communication from trusted sources and improving service delivery design to ensure access in diverse settings and sub-populations, rather than trying to convince those individuals opposed to vaccination.

Our findings underscore that confidence will be a central determinant of dengue vaccine uptake, even in settings with reasonably high vaccine acceptability. Vaccine uncertainty was strongly associated with negative perceptions of the COVID-19 vaccination and with reduced confidence in part due to the rapid development of the COVID-19 vaccines. These results highlight that dengue vaccination programs will need to explicitly acknowledge and address possible residual mistrust stemming from the COVID-19 pandemic, rather than assuming that dengue vaccines would be evaluated by the population independently of this other recent vaccination experience.

Programmatically, our findings imply that communication strategies should emphasize transparency around the dengue vaccine development, safety monitoring, eligibility criteria, number of doses, and expected side effects. This finding aligns with findings from Puerto Rico that found that inconsistent information, concerns about vaccine side effects and risks, and multiple doses were key barriers to Dengvaxia acceptance [67]. Participants emphasized the importance of receiving information from trusted health professionals and institutions, including the Peruvian Ministry of Health, suggesting that integrating dengue vaccination into clinical services – whether facility-based or through outreach to remote communities – may strengthen confidence. Consistently, a multi-site study conducted in Latin America (that included Argentina, Brazil, Colombia, and Mexico) and several Asian countries found higher willingness to vaccinate among individuals who expressed trust in the health system and in their health professionals' ability to manage any side effects [65,66]. Health centers as a source of dengue vaccine information were not significant quantitatively, although this could be due to the phrasing of the question specifying health centers as opposed to healthcare personnel they might know and trust outside

of a clinic. Leveraging the support of locally trusted figures as program champions may further enhance acceptability, particularly when these individuals are visibly vaccinated themselves [68]. These approaches can help normalize the vaccine and reduce possible hesitancy about being among the "first" to receive a new vaccine.

Convenience also emerged as a key and modifiable factor that could shape vaccine uptake. Although participants did not describe long waiting times as a reason to refuse vaccination, various experiences during the COVID-19 vaccination were described as deterrents, including extended lines, limited vaccination hours, and transportation barriers. These findings suggest that dengue vaccination programs should be designed to minimize logistical burdens, particularly for those might accept the vaccination but may delay or miss getting it due to access constraints.

Integrating dengue vaccination into existing outreach platforms may offer an efficient solution that builds on existing infrastructure and services that are already trusted and used. For example, coupling vaccination with established vector control activities or mobile health campaigns could reduce vaccine access barriers while capitalizing on familiar public health infrastructure. Data from Colombia and Singapore found participation in vector control activities was positively associated with dengue vaccine acceptance [66]. Such integration would need to be accompanied by clear messaging that vaccination complements, rather than replaces, vector control efforts, to avoid misconceptions that could undermine other prevention behaviors, as was observed by the questions asked in the FGDs. There must be explicit reinforcement that, despite being vaccinated, environmental management and vector control activities are still critical for overall dengue prevention and control.

Our findings also reveal that vaccine hesitancy is not evenly distributed across the population, suggesting opportunities for targeted interventions. The literature examining the relationship between educational attainment and vaccine acceptance is heterogeneous, with some studies reporting greater hesitancy among individuals with higher education, others identifying higher hesitancy among those with lower education, and several observing no significant association between education and vaccine attitudes [66,69–71]. Our study found individuals with higher levels of formal education were more likely to express uncertainty, underscoring that hesitancy in this context may reflect critical appraisal rather than lack of information (e.g., the possibility that those with higher education in these settings were more influenced by myths that emerged amongst vaccine deniers globally during the pandemic)– or the ability to access services including vaccines from sources this population trusts more, such as the private sector, even if it requires paying. Education level was examined in our quantitative analysis but not during our FGDs, so further qualitative research would be needed to fully explore why this association exists in our population. Similarly, men demonstrated higher levels of uncertainty in crude analyses, which may reflect that women tend to be more regular health consumers (for themselves and children) and hence exposed to messaging at health facilities. This finding reinforces the need for messaging from diverse sources ensure outreach to all subpopulations.

There are a few other take-home issues that warrant discussion. First, during Peru's limited roll-out of the Qdenga vaccine in 2024–25 [72], the dengue vaccine was provided free of charge through MINSA. Our findings suggest that keeping it this way for future rollouts could attract more people who were hesitant for economic reasons; in other Latin American countries, willingness to receive the dengue vaccine was associated with preference for a free vaccine [65,66]. Additionally, even though dengue is endemic in these sites, education about its transmission, clinical symptoms, environmental management, among other topics, must be reinforced. For example, it is well established that second infection with a DENV serotype different from the first infection can progress to a more severe presentation of disease due to antibody-dependent enhancement [73–75]; this is an important message to leverage for individuals who had a mild self-limiting disease to increase their perception of risk.

One potential methodological limitation was our decision to explore attitudes toward a hypothetical dengue vaccine by asking participants to reflect on their decision-making process regarding another recent 'new vaccine'—the COVID-19 vaccine—as a proxy for evaluating acceptance of novel immunization strategies. Peru's COVID-19 vaccination efforts began in February 2021 [76], only a year before our FGDs took place, which likely influenced responses by our participants, as exposure to vaccination information and campaigns were highly publicized and evolving at this time. Some may

argue that the recent COVID-19 experience influenced how people responded to the FG and survey questions, and we fully agree: it is likely to influence many for a long time. But we specifically constructed our FGDs to include people's experience with COVID-19 because we felt it was a real and tangible example of what might drive hesitancy by looking at what happened during the pandemic, rather than discussing an abstract and hypothetical future vaccine. Since this study, there was a limited rollout of the Qdenga vaccine in Iquitos, and we conducted a qualitative study to explore health professionals' and community members' experiences. This manuscript is in preparation, but the findings from this mixed methods study held and are still relevant to future large-scale rollouts.

### Limitations

Several limitations must be considered. First, this study took place in specific urban and peri-urban areas of two dengue endemic sites, which may not reflect views across other geographical and socio-economic settings in Peru or Latin America. Second, this study took place two years after the start of the pandemic, and two years prior to the Qdenga rollout in Peru. Findings reflect attitudes towards a hypothetical dengue vaccine which do not always translate into behaviors. Third, we focused on individuals who were unsure about a vaccination for programmatic reasons; our sample of refusers was small, and we did not explore their perspectives in depth. Future research could explore what is driving the perspectives amongst vaccine refusers. Fourth, self-reported data on vaccine acceptance and hesitancy may be subject to social desirability bias; this may be exacerbated by discussing a dengue vaccine which is not yet available. In other words, the study's focus on attitudes toward a hypothetical dengue vaccine may not capture actual behaviors and decision-making processes once a vaccine becomes available. Finally, almost three quarters of our survey participants were women, which is not representative of the whole population and may affect generalizability of the study, yet in our experience women tend to be the decision makers about health and vaccines in their families in Peru.

### Conclusion

A Future dengue vaccine will provide much needed protection against a disease that is currently relying on one method of prevention: vector control. In dengue endemic settings like Peru, high baseline acceptability for a dengue vaccination offers hope that future rollouts will be well-accepted and successful. However, translating acceptability into sustained uptake will require strategies that prioritize confidence-building, reduce logistical barriers, and strategically target individuals or sub-groups of the population that remain uncertain. By aligning communication with vaccination delivery and targeting these strategies to address community concerns and context, dengue vaccination programs can more effectively complement existing dengue control measures and contribute to reductions of its burden.

### Supporting information

**S1 File. Material: Data collection materials (Focus group guide and survey).**
(PDF)

**S1 Table. Representative Quotes by Themes and Subcodes.**
(DOCX)

**S2 Table. Convergence Matrix of Mixed-Methods Findings.**
(DOCX)

**S1 Text. Variable Selection and Multicollinearity Assessment.**
(DOCX)

**S2 Text. Description and adaptation of Oxford COVID-19 Vaccine Hesitancy Scale.**
(DOCX)

**S3 Text. Description of the development and selection of Dengue Vaccine Hesitancy Outcomes: Analytical strategies used to model dengue vaccine hesitancy.**
(DOCX)

**S4 Text. Sensitivity analysis: Comparison of predictors across outcome specifications.**
(DOCX)

**S5 Text. Description of Methodological Limitations.**
(DOCX)

**S6 Text. Characterization of Participants Excluded from the Multivariable Model due to Missing Covariate Data (Complete-Case Analysis).**
(DOCX)

## Acknowledgments

The researchers would like to thank all the participants in this study as well as the research team members who assisted with survey collection and transcribing the audio.

## Author contributions

**Conceptualization:** Luz M. Moyano, Amy C Morrison, Valerie A. Paz-Soldan.

**Formal analysis:** Jorge L Cañari-Casaño, Emma B Ortega.

**Investigation:** Alfonso S Vizcarra, Roberto Camizan-Castro, E. Jennifer Ríos López, Jhonny J. Córdova López, Cristina Hidalgo, Valerie A. Paz-Soldan.

**Methodology:** Jorge L Cañari-Casaño, Luz M. Moyano, Amy C Morrison, Valerie A. Paz-Soldan.

**Project administration:** Alfonso S Vizcarra, Amy C Morrison, Valerie A. Paz-Soldan.

**Software:** Jorge L Cañari-Casaño.

**Supervision:** Jorge L Cañari-Casaño, Amy C Morrison, Valerie A. Paz-Soldan.

**Validation:** Jorge L Cañari-Casaño, Jhonny J. Córdova López.

**Writing – original draft:** Jorge L Cañari-Casaño, Emma B Ortega.

**Writing – review & editing:** Jorge L Cañari-Casaño, Emma B Ortega, Amy C Morrison, Valerie A. Paz-Soldan.

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
